

**Aerosol chemistry, transport and climatic implications during extreme biomass**
**burning emissions over Indo-Gangetic Plain**
**Nandita Singh[1], Tirthankar Banerjee[1,2], Made P. Raju[3], Karine Deboudt[4], Meytar Sorek-Hamer[5],**
**Ram S. Singh[2,6] and Rajesh K. Mall[1,2]**
[1]Institute of Environment and Sustainable Development, Banaras Hindu University, Varanasi, India
[2]DST-Mahamana Centre of Excellence in Climate Change Research, Banaras Hindu University, Varanasi, India
[3]High Altitude Cloud Physics Laboratory, Indian Institute of Tropical Meteorology, Pune, India
[4]Laboratoire de Physico-Chimie de l'Atmosphère, Université du Littoral Côte d'Opale, Dunkerque, France
[5]NASA Ames Research Center, Moffett Field, CA, USA
[6]Department of Chemical Engineering and Technology, Indian Institute of Technology (BHU), Varanasi, India
*Correspondence to*: Tirthankar Banerjee (tb.iesd@bhu.ac.in; tirthankaronline@gmail.com)
**Abstract**
The large-scale emissions of airborne particulates from burning of agricultural residues particularly
over the upper Indo-Gangetic Plain (IGP) have often been associated with frequent formation of haze,
adverse health impacts, modification in aerosol climatology and thereby aerosols impact on regional
climate. In this study, short-term variations in aerosol climatology during extreme biomass burning
emissions over IGP, and thereby to regional climate were investigated. Size-segregated particulate
concentration was initially measured and submicron particles ($PM_{1.1}$) were found to dominate
particulate mass within the fine mode ($PM_{2.1}$). Particulate bound water-soluble ions were mainly
secondary in nature, primarily composed of sulfate and nitrate. There was evidence of gaseous $NH_3$
dominating neutralization of acidic aerosol species ($SO_4^{2-}$) in submicron particles, in contrast to crustal
dominating neutralization in coarser particulates. Variation in black carbon mass ratio was found to
be influenced by local sources, while sudden increase in concentration was consistent with high Delta-
C, referring to biogenic emissions. Influence of biomass burning emissions were established using
specific organic (levoglucosan), inorganic ($K^+$ and $NH_4^+$) and satellite (UV Aerosol Index, UVAI) tracers.
Levoglucosan was the most abundant within submicron particles (649±177 ng m$^{-3}$), with a very high
ratio (>50) against other anhydrosugars, indicating exclusive emissions from burning of agriculture
residues. Temporal variations of all the tracers were consistent, while $NH_4^+$ was more closely
associated to levoglucosan. Spatio-temporal distribution of aerosol and few trace gases (CO and $NO_2$)
were evaluated using both space-borne active and passive sensors, and a significant increase in
columnar aerosol loading (AOD: 0.98) was evident during extreme biomass burning emissions, with
presence of absorbing aerosols (UVAI > 1.5) having low aerosol layer height (~1.5 km). A strong
intraseasonality in aerosol cross-sectional altitudinal profile was even noted from CALIPSO, referring
dominance of smoke and polluted continental aerosols across IGP. Possible transport mechanism of
biomass smoke was established using cluster analysis and concentration weighted of air mass back-
trajectories. Short-wave aerosol radiative forcing (ARF) was further simulated considering
intraseasonality in aerosol properties, which resulted in considerable increase of atmospheric ARF
(135 Wm$^{-2}$) and heating rate (4.3 K day$^{-1}$) during extreme biomass burning emissions compared to non-
dominating one (56 W m$^{-2}$, 1.8 K day$^{-1}$). We therefore conclude that influence of biomass burning
emissions on regional aerosol climatology must need to be studied in much finer scale to improve
parameterization of aerosol/-climate model across the region.



## 1. Introduction

Aerosols are studied systematically in terms of their potential to influence the transfer of radiant energy and distribution of latent heat, by which it modifies the Earth's weather and climate. Aerosols are also associated with nutrient recycling and for governing atmospheric chemistry (Kanakidou et al., 2018). Aerosol interaction with radiation mainly constitutes its radiative forcing of climate change (Bellouin et al., 2005; Bond et al., 2013) while, it also modifies the climate by means of cloud formation processes (Seinfeld et al., 2016). The aerosol-radiation interaction necessitates understanding on spectrally varying aerosol optical properties, which are associated to particle size distribution, chemical composition, morphology and mixing states. The representation of aerosol processes in global/-regional climate models varies considerably and thereby, estimates of aerosol-radiation interaction still consist significant level of uncertainties (Myhre et al., 2013). This necessitates extensive regional investigation in terms of aerosol composition and properties for improved parametrization of aerosol schemes in the regional/-global climate model.

The Indo-Gangetic plain (IGP) in South Asia is especially unique in terms of aerosols loading and diversity that varies over the seasons (Singh et al., 2017a,b; Sen et al., 2017; Sayer et al., 2014; Kumar et al., 2018). The IGP is often projected to be one of the most vulnerable region in terms of aerosol induced negative health impacts (Apte et al., 2015) and therefore, numerous observational and modeling studies were made for better characterization of aerosols (Sen et al., 2017; Moorthy et al., 2008 and references therein). Recently, Singh et al. (2017a) has concluded the presence of considerable spatial and seasonal variations in aerosol sources over South Asia, with vehicular emissions, followed by industrial emissions and secondary aerosols contributing most of the anthropogenic emissions of fine particulates. However, episodes of specific emissions like from biomass burning (Wan et al., 2017; Rajput et al., 2011, 2014; Rajput and Sarin, 2014) and use of fire crackers (Kumar et al., 2016) also induce sudden large-scale changes in aerosol properties, and necessitate extensive investigation for better representation in regional aerosol model. Post-harvest agricultural residue burning, especially over upper IGP is projected to release 400 Gg of particulate bound organic aerosols (OA) and 40 Gg of black carbon (BC, Rajput et al., 2014), almost entirely (90 %) from burning of rice husks (Rajput et al., 2011). The OA mostly constitute the fine particulate mass (20-90 %) and are reported to be hydrophilic in nature (Rajput and Sarin, 2014) therefore, pose potential to act as CCN molecule, or at most compete with sulphate particle (Singh et al., 2017b). Nevertheless, presence of such huge amount of OA may either lead to reduction in mean evaporation and modify regional precipitation or may reduce cloud formation processes by inducing additional heat to the system (Riipinen et al., 2011; Sun and Arriya, 2006). The biomass burning aerosols also impact the Earth's surface albedo by depositing on glaciers. The net radiative forcing of biomass



burning aerosols by aerosol–radiation interactions is close to neutral i.e. - 0.0 (-0.20 to + 0.20) W m$^{-2}$,
having a gradient with negative forcing from OA and positive forcing from BC (Myhre et al., 2013).
Biomass burning aerosols even evolve due to oxidation (Jimenez et al., 2009; Vakkari et al., 2014),
from gas-phase precursors to semi-volatile secondary OA (SOA) and finally to highly volatile oxidized
gases (e.g. CO and $CO_2$), thus warrants molecular characterization and specific understanding both in
terms of composition, atmospheric chemistry, transport and radiative forcing (Singh et al., 2017b).
Several investigations were made to understand the characteristics of biomass burning
aerosols exclusively over IGP. Few attempts were made solely using ground-based information e.g.
aerosol emission budget (Rajput et al., 2014), organic mass-to-organic carbon ratio (Rajput and Sarin,
2014), emissions of PAHs (Rajput et al., 2011), organic molecular tracers (Wan et al., 2017) and
radiative forcing (Sharma et al., 2017; Alam et al., 2011); while few have explored remote sensing
observations to interpret fire (Vadrevu et al., 2012) and aerosol plume characteristics (kaskaoutis et
al., 2014). However, there is a need to integrate both ground and contemporary satellite-based
information so that spatio-temporal characterization of aerosols and its climatic impacts are assessed
more realistically. In the present analysis complementary measurements from both ground and space-
based platforms are therefore combined to trace the vital signatures of extreme biomass burning
emissions, its chemical evolution, transport and aerosol radiative forcing. Initially, chemical
speciations of size-segregated aerosols are made, supported by black carbon dynamics, molecular
tracers of biomass emissions, and further explored in terms of their relevance to regional
meteorology. The spatial extent of aerosol emission and transport was made using Modern-Era
Retrospective Analysis for Research and Applications (MERRA) atmospheric reanalysis data, Global
Data Assimilation System (GDAS) archives and NCEP/ NCAR Reanalysis data. Further, visualization
from 'A-Train' satellite constellation, from both space-borne passive sensors like MODerate resolution
Imaging Spectroradiometer (MODIS), Ozone Monitoring Instrument (OMI) and active sensor like
Cloud-Aerosol Lidar and Infrared Pathfinder Satellite Observation (CALIPSO) are included. Briefly, the
results are explored to highlight three exclusive but inter-related mechanisms, i.e. aerosol chemistry,
regional transport and radiative forcing, and their intra-seasonal variations over middle IGP, which
may well be useful for improving aerosol scheme in regional climate model.
**2 Experimental methods**
**2.1 Site description**
Ground-based aerosol measurements were made at the institutional premises of Banaras
Hindu University, Varanasi (25.26 °N, 82.98 °E, 82 m AMSL). The ground station typically experiences
a humid sub-tropical climate, with no localized effects of oceans or mountains (Fig. 1). The



predominating wind profile is north-westerly which are projected to subsidize over a section of middle
IGP, coinciding well with the ground monitoring station, thereby facilitates gradual accumulation of
aerosols (Kumar et al., 2018). Interestingly enough, the region also experiences a significant diurnal
variation in atmospheric boundary layer (ABL) associated with high convective turbulence that usually
redistributes aerosols to a greater altitude (Kumar et al., 2015, 2017a). Particulates emitted from
crustal sources, road dust re-suspension, vehicular exhausts and biomass/waste burning are often
reported to primarily constitute the regional aerosol climatology (Singh et al., 2017a).
**2.2 Micro-meteorology, ABL and wind field**
The 24 h average meteorological parameters e.g. temperature, relative humidity (RH) and
wind speed (WS) were obtained from wunderground.com and validated with regional weather
monitoring station data. The ABL heights at specific coordinate were retrieved from Global Data
Assimilation System (GDAS) archives hosted at NOAA-Air Resource Laboratory, which provides
simulated meteorological observations at a gridded scale. The 3-hourly ABL data (0.5°) were averaged
on daily basis in parallel to period of particulate measurement. The NCEP/NCAR Reanalysis data was
used to measure the variation of 3-D wind fields at near surface (1000 m) with a horizontal resolution
of 2.5° X 2.5°. Vector wind composite mean (m s$^{-1}$) for 925 hPa was plotted for the defined coordinate
(6-38 °N, 50°-105 °E) to understand the synoptic pattern of wind field.
**2.3 Ground-based measurements**
2.3.1 Size-segregated aerosol mass concentration
Size-segregated aerosols were collected on pre-combusted quartz fiber filter using Anderson
eight-stage cascade impactor (Tisch Environmental Inc., USA). Sampling was continued for once in a
week from 1$^{st}$ October to 15$^{th}$ December 2016, continuously for 72 h (in each week) to get
representative deposition of particulates. The instrument was run with a fix flow rate of 28.3 LPM,
having aerodynamic cut-off diameter of <0.43, 0.65, 1.1, 2.1, 3.3, 4.7, 5.8 and >9.0 μm (with 50 %
collection efficiency). The individual stages of each sample were then segregated into three groups on
the basis of cut-off diameter (i) coarse mode (PM$_{>2.1}$) comprising the stages with the aerodynamic
diameter >2.1 μm; (ii) fine mode (PM$_{1.1-2.1}$) for the stages with diameter 1.1 to 2.1 μm; and, (iii)
submicron mode (PM$_{<1.1}$) for the last two stages with the diameter <1.1 μm.
2.3.2 Black carbon mass concentration
The black carbon (BC) real-time mass concentration was measured using a seven channel
Aethalometer (Model AE-42; Magee Sci. Inc., USA), with a constant flow rate of 3 LPM at 5 minutes
resolution. Aethalometer measures the attenuated beam of light transmitted through aerosol sample



on filter tape at seven wavelengths (370, 470, 520, 590, 660, 880 and 950 nm), while attenuation at
880 nm was considered for BC (Bodhaine, 1995). The BC concentration is estimated based on the
concept of linearity between the light attenuation and BC mass deposited on quartz filter. An
absorption efficiency of 16.6 $m^2\,g^{-1}$ (provided by the manufacturer) was used to measure BC after
correction of loading effect. The detailed mechanism for estimation of BC is described in Wang et al.
(2011) and Kumar et al. (2017a). Additionally, BC measured at two wavelengths e.g. 370 nm (indicating
absorption by wood-smoke particles) and 880 nm (by both fossil fuel and wood burning emissions)
were used to compute Delta-C ($BC_{370nm} - BC_{880nm}$). Delta-C is reported to symbolize smoke emissions
(Wang et al., 2011; Kumar et al., 2016) and therefore, was used as a tracer for biomass emissions.
2.3.3 Aerosol chemical constituents
*Water-soluble ions*
The particulate deposits on filter were extracted with deionized water in an ultrasonic bath
(Microclean-109, Oscar, India) for 30 min, and extracts were further filtered through syringe filters
(pore size 0.2 μm). The water-soluble ionic constituents (WSIC) were analyzed by ion exchange
chromatograph (ICS 3000, Dionex, USA). For measurement of anions ($Cl^-$, $NO_3^-$, $SO_4^{2-}$ and $PO_4^{3-}$), the IC
was equipped with a micro-membrane suppressor (AERS-300, 4 mm; Dionex) with IonPac analytical
column (AS11-HC × 250-mm) connected with a guard column IonPac (AG11-HC, 4×50mm; Dionex).
Cations ($NH_4^+$, $Na^+$, $K^+$, $Mg^{2+}$, $Ca^{2+}$) were measured through a suppressor (CERS-300, 4 mm; Dionex)
with an analytical column (IonPac CS12A-HC, 4×250 mm; Dionex) and a guard column (IonPac CG11-
HC, 4 × 50 mm; Dionex, USA). The background contamination was removed by subtracting the blank
filter value from sample values (Kumar et al., 2017b).
*Trace metals*
The trace metals were extracted from filter discs as per US EPA Method IO-3.2 (EPA, 1999).
The filters were cut into pieces and digested in acid mixture solution (5.55 % $HNO_3$ with 16.67 % HCl)
on a hot plate for 2 h. The extracts were filtered, stored at 4 °C and were analyzed by atomic
absorption spectrophotometer (Avanta Ver 2.01, GBC) for Cu, Mn, Fe, Cd, Cr, Pb, Ni, Co, and Zn.
*Organic compounds*
For determining the aerosol organic constituents, the filter composites of each group were
extracted by ultrasonicating the filters initially with dichloromethane-hexane mixture (1:1), followed
by dichloromethane-methanol mixture (1:1). Both solvent extracts were combined and concentrated
using vacuum rotatory evaporator and nitrogen evaporator to a volume of 100 μL (Hu et al., 2013).
The extracts were derivatized by silylation with N, O-bis-(trimethylsilyl)-trifluoroacetamide and 1 %



trimethylchlorosilane prior to analysis. After derivatization, the residue was re-dissolved in hexane
and analyzed by gas chromatography-mass spectrometry (GCMS-QP2010 Ultra, Shimadzu, Japan)
equipped with Rxi-5MS fused silica capillary column having dimension 30 m x 0.25 mm id x 0.25 μm
(Restek, Bellefonte, PA, USA). The 1 μL of samples were injected in GCMS at 260°C injector
temperature in splitless mode. The column oven temperature program was started at 50°C with 2 min
of the isothermal hold which further raised up to 120 °C (linear elevation @ 30 °C min$^{-1}$) and 300 °C
(linear elevation @ 6 °C min$^{-1}$) followed by the isothermal hold of 11 min. The electron impact
ionization was used to produce molecular ions at 70 eV with the ion source and interface temperature
of 230 °C and 270 °C, respectively. The molecular ions were scanned for a wide range of m/z from 40
to 650. The target compounds were identified on the basis of retention time and fragmentation
pattern from National Institute of Standards and Technology (NIST) library and standard solutions of
analytes.
**2.4 Satellite-based observations**
**2.4.1 Aqua-/Terra MODIS data**

15       The aerosol optical depth (AOD) at 550 nm was retrieved daily from MODIS onboard Aqua

satellite in parallel to ground-based aerosol monitoring. The level 2 Collection 6 AOD at 10 km
resolution was retrieved using MODIS merged DT-DB AOD (AOD_550_-
Dark_Target_Deep_Blue_Combined, Levy et al., 2013). The selection of merged DT-DB for retrieving
AOD was based on higher retrieval number and accuracy across the IGP (Mhawish et al., 2017). The
AOD for the ground station was calculated as the average of 5 x 5 pixels, around the monitoring site.
Angstrom exponent (AE, α) was retrieved using MODIS C6 level 2 DB AOD and relation between AOD
and AE was used to measure the aerosol loading and the particle size (Kumar et al., 2015; Mhawish et
al., 2017). Columnar water vapor content (CWV) was retrieved from Aqua MODIS collection 6 level 2
infrared channel at 1km spatial resolution. To illustrate the impact of biomass burning, the fire spots
were retrieved over the IGP from Aqua-/Terra MODIS Fire Mapper product (collection 6, spatial
resolution 1×1 km$^2$) provided by the Fire Information for Resource Management System (FIRMS,
https://firms.modaps.eosdis.nasa.gov). The details about MODIS fire products and its algorithm may
be found elsewhere (Justice et al., 2006).
**2.4.2 Aura-OMI and MERRA-2 reanalysis data**

30       The OMI onboard AURA satellite has a typical daily global coverage with 13 × 24 km$^2$ spatial

resolution at nadir and measures solar backscatter irradiation in the UV–visible spectrum (264-504
nm; Levelt et al., 2006). Ultraviolet Aerosol Index (UVAI), tropospheric NO$_2$, total columnar ozone
(TCO) and Single scattering albedo (SSA) were retrieved from Aura OMI available at NASA Goddard



Earth Sciences Data and Information Services Centre (GES DISC). Aura OMI UVAI is capable of detecting
aerosol absorption from satellite measured radiances without any prior assumption on aerosol
composition (Torres et al., 2013). It is a qualitative parameter and is widely used to identify the UV
absorbing aerosols (e.g. smoke plumes, soot and mineral dust; Torres et al., 2013; Mhawish et al.,
2018). The UVAI based on OMI near-UV aerosol retrieval algorithm (OMAERUV) was extracted from
Level 2G, version 003 aerosol product containing one day's Level 2 data set of original pixels (13 × 24
$km^2$) into 0.25° x 0.25° grids. The $NO_2$ tropospheric column density was retrieved from cloud screened
(cloud fraction <30 %) Level 3, version 003, daily 0.25° x 0.25° gridded OMNO2d product (Krotkov et
al., 2017). To estimate TCO, Level 3e data (OMDOAO3) at a spatial resolution of 0.25° x 0.25° was used.
SSA at 550 nm was retrieved from OMI level 2G product (OMAERUV) at 0.25° × 0.25° resolution. The
Carbon Monoxide (CO) surface concentration (in ppbv) was retrieved from Modern-Era Retrospective
Analysis for Research and Applications, version 2 (MERRA-2) atmospheric reanalysis data available at
0.5° x 0.625° from GES DISC.
**2.4.3 CALIPSO-CALIOP observations**
CALIPSO products were used to examine the vertical distribution of aerosols, altitude of
aerosol layers, clouds, aerosol types and their properties at visible (532nm) and near-IR wavelengths
(1064 nm). The V4.10 CALIOP Level 2 altitude-orbit cross-section profiles obtained from CALIPSO sub-
setting web application (https://www-calipso.larc.nasa.gov) was used. The Lidar profiles were
processed for images of vertical feature masks, aerosol subtypes and extinction coefficients (at 532
nm) at 30 m vertical resolution over the selected grid (80°-86 °N and 22°-28 °E). The details about data
products, calibration and uncertainty are discussed in Rogers et al. (2011).
**2.5 Air-mass back trajectory**
The NOAA HYSPLIT model (Draxler and Rolph, 2003) was used to simulate particle back
trajectories in a three-dimensional system. The HYSPLIT was run on using the Global Data Assimilation
System data (GDAS, 0.5°×0.5°) available from archive dataset (http://ready.arl.noaa.gov/gdas1.php)
to predict 120 h air-mass back trajectories (00:00, 06:00, 12:00 and 18:00 UTC) starting from October
to December 2016. Trajectories for different aerosol loading periods were then overlaid on MODIS
fire map to study the transboundary movement of emissions from biomass burning. The trajectory
analysis was made using GIS-based software TrajStat (Wang et al., 2009). Concentration weighted
trajectories (CWT) were also drawn considering columnar aerosol loading to evaluate potential
aerosol source fields and mechanism of aerosol transport over the Gangetic plain. The specificities of
the models' parameters and algorithms are detailed elsewhere (Wang et al., 2009; Kumar et al., 2018).





**2.6 Aerosol optical properties, radiative forcing and heating rate**
Aerosol induced shortwave (0.2–4.0 µm) direct radiative forcing (ARF) was estimated using
Santa Barbara DISORT Atmospheric Radiative Transfer (SBDART) model (Ricchiazzi et al., 1998). The
SBDART estimates plane-parallel radiative transfer in a clear sky condition for both Earth's top of the
atmosphere (TOA) and at the surface (SUF), while atmospheric forcing (ATM) is calculated as the
difference between them. The standard atmospheric profile is used together with input variables e.g.
AOD, SSA, CWV, TCO and asymmetry parameter (ASP) derived through OPAC model (Optical
Properties of Aerosols and Clouds; Hess et al., 1998). The OPAC provides aerosol optical properties
over a wide range of wavelength and delivers necessary input to SBDART. Mean mass concentrations
of aerosol water soluble (WSIC) and insoluble (dust and organics) components along with BC mass
concentrations were converted to particle number densities and introduced to OPAC for deriving
aerosol optical properties. The OPAC derived outputs were tuned in respect to measured relative
humidity. The AOD and SSA were further reconstructed to match modelled and satellite derived values
within ±5 % deviation.
The weekly mean values of AOD, SSA, ASP, CWV, TCO, visibility and AE were included as an
input to SBDART. The SBDART includes multiple scattering in a vertically inhomogeneous, non-
isothermal plane-parallel media, and is reported to be efficient in resolving the radiative transfer
equation (Raju et al., 2016). The ARF was calculated using 10 solar zenith angles (0 to 89, with
increment of 10) and proceed for conditions like 'with aerosols' or 'without aerosols'. The surface
albedo was decided based on visual observation considering a combination of snow, ocean, sand and
vegetation. Overall uncertainty in the estimated ARF was in the range of 10–15 % (Alam et al., 2011).
The ATM-ARF was further used to compute aerosol atmospheric heating rate ($\partial T/\partial t$, K day$^{-1}$), using
equation (1):
$\partial T/\partial t = (g/Cp)*(\Delta F/\Delta P)$            (1)
where $\Delta P$ is the difference in forcing, $\Delta P$ is the pressure difference between top and bottom boundary
layer, $Cp$ is specific heat capacity of air at constant pressure and $g$ is the acceleration due to gravity
(Kumar et al., 2017a).
**3. Results and discussion**
**3.1 General characteristics of aerosols**
The weekly variation in particulate concentrations in different size fractions are presented in
Fig. 2 with the descriptive statistics in Table S1. The total aerosol mass concentrations have high intra-
seasonal variations (median: 370; range: 134-734 µg m$^{-3}$), mainly influenced by coarse mode particles





(PM$_{>2.1}$) contributing 63±15 % of particulate mass. In contrast, contribution of submicron (PM$_{<1.1}$:
27±12 %) and fine mode particles (PM$_{1.1-2.1}$: 10±4 %) to total aerosol loading were relatively less
(<37%). The average (±1σ) mass concentration of PM$_{2.1}$ (PM$_{<1.1}$ + PM$_{1.1-2.1}$) and total aerosol loading
was 162 (±123) and 390 (±199) µg m$^{-3}$, which were approximately 98 % (against PM$_{2.5}$) and 92 % higher
compared to annual averages observed over the monitoring station (Murari et al., 2017; Prajapati and
Tripathi, 2008). To our knowledge, there are no published reports on submicron particle concentration
over the ground station. Time-series analysis of size-segregated particulates (Fig. 2) indicate the
submicron (PM$_{<1.1}$) and fine mode particles (PM$_{1.1-2.1}$) only had a late rise in mass concentrations, while
the coarse mode particulates (PM$_{>2.1}$) did not show any trend. However, there was a definite increasing
pattern in fine to coarse particle ratio (PM$_{2.1}$/PM$_{>2.1}$; mean: 0.7±0.5; range: 0.2-1.5), due to a
continuous increase of the fine mode from mid-November to the end of the monitoring. Thus the
contribution of fine mode particle to total aerosol loading increased from mid-November (>40 %), and
contributed almost 60 % of particulate mass during the month of December. The submicron particles
also indicate a high median concentration (96 µg m$^{-3}$) compared to fine mode (33 µg m$^{-3}$), and the
particle ratio (PM$_{<1.1}$/PM$_{1.1-2.1}$) remain >1 throughout, only to exceed values >2.5 from November to
December. This clearly indicates the dominance of submicron particles within fine mode fractions,
possibly associated to anthropogenic emissions, and also influenced by local meteorological
conditions e.g. low temperature (mean±SD: 20±3 °C), calm wind (mean: 0.6 m s$^{-1}$) and shallow
boundary layer height (mean±SD: 379±89 m).
**3.2 Aerosol chemical speciations**
*3.2.1 Water soluble inorganic species (WSIS)*
Temporal variation of WSIS in size-segregated airborne particulates are presented in Fig. 3a.
It indicates the major contribution of WSIS to submicron (21 %) and fine particle mass (21 %) compared
to coarser particles (13 %). The secondary inorganic aerosols (SIA = SO$_4^{2-}$ + NO$_3^-$ + NH$_4^+$) together
accounted for 17 % of the submicron particle mass, with major contributions from sulfate (9%) and
nitrate (6 %). Similar was the case for fine particulates as SIA contributed to almost 17 % of aerosol
mass with predominate contribution from sulfate (8 %) and nitrate (6 %), and a relatively small
proportion of ammonia (4 %). In contrast, the relative contribution of SIA to coarse particulate was
lower (7 %), also primarily associated to sulfate (5 %) and nitrate compounds (2 %). This indicates the
secondary nature of origin of fine and submicron particles which possibly evolve through gas-phase
photochemical conversion of SO$_2$ and NO$_2$, eventually neutralized by crustal species like carbonate
salts (CaCO$_3$ and MgCO$_3$) associated with the airborne dust (Murari et al., 2015, 2016). The time-series
of SIA contribution to particulate mass (Fig. 3a) indicate a dominance (although in different extent) of
secondary aerosols in PM$_{1.1-2.1}$ and PM$_{>2.1}$ only during November.





Among the WSIS, $SO_4^{2-}$ was invariably the most abundant within each particulate size fraction
($PM_{<1.1}$: 39 %, $PM_{1.1-2.1}$: 32 %, $PM_{>2.1}$: 36 %), followed by $NO_3^-$ ($PM_{<1.1}$: 27 %, $PM_{1.1-2.1}$: 29 %, $PM_{>2.1}$: 17
%). The $NO_3^-$ / $SO_4^{2-}$ ratio was considered as an indicator of the mobile and stationary sources of
nitrogen and sulfur (Tian et al., 2016). An average ratio varying from 0.62 to 1.92 was noted for all
size-segregated particulates testifying dominance of both sources, although in different time-scales.
In later phase, the ionic ratio ($NO_3^-$ / $SO_4^{2-}$) enhanced (>1) in submicron and fine mode particles, well
identical to the reported haze events over Guangzhou (Tan et al., 2009) and Suzhou, China (Tian et al.,
2016). A very high $NO_3^-$ / $SO_4^{2-}$ ratio (3.2±1.3) was only noted in fine aerosols during October, mainly
due to lower concentration of sulphate. The next two dominant contributors to WSIS were $NH_4^+$
($PM_{<1.1}$: 14 %, $PM_{1.1-2.1}$: 19 %, $PM_{>2.1}$: 5 %) and $K^+$ ($PM_{<1.1}$: 8 %, $PM_{1.1-2.1}$: 5 %, $PM_{>2.1}$: 2 %), both considered
as a molecular tracer for biogenic emission (Banerjee et al., 2015). They constitute the greater
proportion of WSIS in $PM_{<1.1}$ and $PM_{1.1-2.1}$, especially from last week of October till the end of
November, signifying elevated contribution of biomass/ agro-residue burning emissions to these
particle sizes. Further, a strong correlation ($R^2$=0.9) between $NH_4^+$ and $SO_4^{2-}$ and high $NH_4^+/SO_4^{2-}$
equivalent ratio (0.9±0.2) in submicron particulates indicate the abundance of gaseous $NH_3$ to
neutralize acidic species ($SO_4^{2-}$) by forming $(NH_4)_2SO_4$ and/or $NH_4HSO_4$. The $NH_4^+/SO_4^{2-}$ equivalent ratio
gradually increased from week 5 (mean: 1.2, range: 0.9-1.3), possibly due to abundant emission of
$NH_4^+$ from biomass emissions. Unlike submicron particles, the low $NH_4^+/SO_4^{2-}$ equivalent ratios (<0.7,
mean: 0.4) in coarse mode particles indicate the predominant neutralization by crustal minerals
instead of $NH_3$.
Unlike the other WSIS, $Na^+$ and $Ca^{2+}$ were found to contribute maximum in $PM_{>2.1}$ ($Na^+$: 2 %;
$Ca^{2+}$: 3 %), referring their crustal origin. The relative abundance of $Cl^-$ in size-segregated aerosols was
roughly equal for each size fraction, contributing almost in identical to total WSIS in $PM_{<1.1}$ (6 %), $PM_{1.1-}$
$_{2.1}$ (5 %) and $PM_{>2.1}$ (4 %). The possible origin of $Cl^-$ in $PM_{>2.1}$ could be the aged sea salt, transported
from Bay of Bengal, but its association with $PM_{<1.1}$ was most likely due to biomass burning emissions
(Pavuluri at al., 2011; Murari et al., 2015). The temporal variations of WSIS in all particulate size
fractions were consistent except for $Mg^{2+}$ and $PO_4^{3-}$ with a contribution of both ions to particulate
masses less than 0.2 %, indicating their non-biomass specific emission. A strong correlation was noted
between the anion and cation equivalents within all the groups (0.7-0.9) indicating that maximum ions
were from the filter samples. The total ion equivalent ratio (anions to cation) refer a cationic imbalance
($PM_{<1.1}$: 1.2, $PM_{1.1-2.1}$: 0.8 and $PM_{>2.1}$: 0.6) with excess cations in fine and coarse mode particles,
possibly due to unmeasured components like carbonates and bicarbonates.
*3.2.2 Trace metals*



Total metallic contribution to particulate mass was found maximum in $PM_{1.1-2.1}$ (24 %),
followed by $PM_{>2.1}$ (11 %) and least in $PM_{<1.1}$ (7 %, Fig. 3b). The most abundant elements were Na, Ca,
K and Zn for all size fractions, contributing 90-98 % of total identified metals, while the remaining
fractions were primarily constituted by Fe (1-10 %). Within the detectable level of metals, Ca and Na
share 88 % of metal concentrations in $PM_{<1.1}$, contributing 7% of submicron particulate mass, without
having any specific temporal trend.  However, Ca, Na were found high in $PM_{1.1-2.1}$ (Ca: 10 %; Na: 7 %),
referring their origin from resuspension of crustal materials and road dust.
There are some evidences of trace metal emissions from burning of biomass. Wang et al.
(2015) have concluded biomass combustion as the most prominent source of Fe concentration for
submicron particles. For this analysis, although Fe was measured maximum in $PM_{>2.1}$, the relative
increase in Fe concentration in submicron ($PM_{1.1}$: 59 %) and fine aerosols ($PM_{1.1-2.1}$: 415 %) during week
6 to week 9 possibly indicate the added contribution of biomass burning emissions.  Beside Fe, there
are also reports of trace metals emissions particularly K, Cu, S, Zn, Pb from burning of rice-straw (Ryu
et al., 2012); organic bound $Fe^{2+}$, $Cu^{2+}$, $Ni^{2+}$, $Zn^{2+}$ from hardwood burning (Chang-Graham et al., 2011)
and Cu, Pb, Ni, As from the burning of biomass fuel (Zhang 2014). In our case, massive increase in K
($PM_{1.1}$: 528 %; $PM_{1.1-2.1}$: 119 %) was noted between week 6 and week 9. This contrasted with coarse
particle bound Fe and K which are primarily of crustal origin (Banerjee et al., 2015), and recorded 15
% (Fe) and 83 % (K) increase in concentration within week 6 to 9. Zinc was found considerably high in
$PM_{>2.1}$ (3 %) and relatively small proportion in $PM_{1.1-2.1}$ (2 %). The major sources of atmospheric Zn are
burning of residual oil, refuse and garbage (Gonzalez et al., 2016) which possibly leads to higher mass
fractions in coarser particulates. Even, a relatively high Zn concentration was noted irrespective of
particulate size in later phase of monitoring coincide with the winter specific burning of waste/- refuse
over the region (Kumar et al., 2017b). The relative contribution of rest of the trace metals (e.g. Mn,
Pb, Cd, Ni, Cu, Cr and Co) in particulate mass were insignificant (<0.05 %), without having any specific
temporal pattern.
**3.3 Characteristics of BC mass loading**
Daily average BC concentration and Delta-C ($BC_{370} - BC_{880}$) are plotted in Fig. 4 with some data
gaps. The 24 h average BC concentration varied from 2.0-15.4 µg m$^{-3}$ with a seasonal mean (±1σ) of
8.3 (±2.9) µg m$^{-3}$. The season specific BC average was 80 % higher in comparison to annual mean
reported over the ground station (4.6 µg m$^{-3}$; Kumar et al., 2017a), while there are also reports of
winter-specific very high BC mixing ratio (22 µg m$^{-3}$; Murari et al., 2016) that usually persist over the
region. Over middle IGP, vehicular exhausts mainly regulate the BC profile (Kumar et al., 2016, 2017a;
Murari et al., 2016), while isolated cases like large-scale burning of agriculture residues/ biomass/
waste and emissions from residential heating also contribute in BC concentrations (Kumar et al.,





2017b). The hourly BC profile indicates a distinct diurnal profile with a general trend of high
concentration (>9 µg m$^{-3}$) during night to early morning hours (21:00-7:00 h), and a low concentration
(<6 µg m$^{-3}$) during day time (11:00-17:00 h). A gradual rise in BC in morning and evening hours well
coincide with the traffic rush hours, indicating the contributions of traffic emissions. However, the rest
of BC diurnal profile is mainly attributed to the variation in boundary layer height, which otherwise
considered as the most important factor in regulating BC after the source itself (Kumar et al., 2016,
2017a).

8         The temporal plot shows an enhanced BC concentration from the end of October (week 4) till

the November end (week 9) possibly due to increased source strength, in addition to the influence of
local meteorology. While there was no significant variation in meteorological variables within this
timeframe (Table S1), we hypothesize that the variation in BC source strength might have well
influenced the BC concentration. To understand the variation in BC sources, 24 h average Delta-C
concentration (mean±SD: 2.3±1.0 µg m$^{-3}$) is also included in Fig. 4, which refers a temporal shift in BC
sources. Except few exceptions, high Delta-C (>2.3 µg m$^{-3}$) was observed particularly in the month of
November (80 % of days) and December (46 %), referring added contribution of biomass burning
emissions.
**3.4 Composition of organic aerosols**

18        Size-segregated particle-bound organic aerosols (OA) were analysed for 22 *n*-alkanes (C$_{13}$-C$_{34}$),

3 anhydrosugars (levoglucosan, mannosan and galactosan), 4 PAHs and 10 *n*-alkanoic acids (C$_{12}$-C$_{26}$)
(Fig. 5). Considerable variation in the concentration and size distributions of these OA were
noted. Contributions of OA to size-segregated particulates were relatively less because of partial
characterization through GC-MS. Among the identified species, *n*-alkanes were invariably the highest
within PM$_{<1.1}$ (mean±SD: 484±103 ng m$^{-3}$) compared to fine (267±43 ng m$^{-3}$) and coarse mode aerosols
(308±93 ng m$^{-3}$). The molecular distribution of *n*-alkanes homologues in all three size fractions showed
a slight dominance of odd-numbered *n*-alkanes. The CPI (Carbon Preference Index) remain close to
unity (CPI range: 1.2-2.1; mean±SD: 1.5±0.5), indicating dominance of anthropogenic emissions like
combustion of fossil fuels and biomass burning. The higher molecular weight homologues (>C$_{25}$)
concentration were found highest in PM$_{<1.1}$ with an oscillating pattern, having odd molecules
concentration higher than the adjacent even molecules (Fig. 5b). In contrast the low molecular weight
homologues (<C$_{25}$) showed no such specific pattern of odd/even dominance. The sources of higher
homologues (C$_{27}$, C$_{29}$ and C$_{31}$) are probably the surface deposited plant litter for coarse mode and
biomass burning for fine mode aerosols, while low molecular weight homologues (<C$_{25}$) primarily
originate from the fossil fuel combustion (Kang et al., 2016). Saturated fatty acids were found to
constitute a larger fraction of solvent extractable organics within coarse mode (439±38 ng m$^{-3}$)



and submicron particles (357±162 ng m$^{-3}$) in comparison to fine mode (171±57 ng m$^{-3}$). For all three
size fractions, total low molecular weight fatty acids (≤C$_{20}$) concentration was found higher than the
high molecular weight fatty acids (≥C$_{20}$), indicating the anthropogenic emissions like vehicular,
residential biomass burning and energy practices. Presence of high concentration of C$_{12}$, and C$_{15}$ refer
the dominance of cooking oil combustion. The high concentration of C$_{22}$ further suggests the influence
of biomass burning which potentially emit both, high and low fatty acids (Mochida et al., 2007). The
fatty acid amide was found in trace amount which could possibly be derived from fatty acid and
ammonia during burning process. Presence of PAHs were comparatively less, primarily within
submicron particles which were mainly emitted from incomplete combustion of wood, oils and coal.
Levoglucosan was found to be the most abundant in the submicron particles with an average
(±1σ) of 649 (±177) ng m$^{-3}$. In contrast, concentration in fine (229±87 ng m$^{-3}$) and coarse particles
(162±68 ng m$^{-3}$) levoglucosan concentrations were relatively low, referring the dominating influence
of burning emissions in submicron particles. Levoglucosan concentration measured in this study are
well comparable to other reported observations, especially with the cases that have accounted the
influence of biomass burning emissions e.g. New Delhi (1978 ng m$^{-3}$, Li et al., 2014), Mt. Tai, China (391
ng m$^{-3}$, Fu et al., 2008), Gent, Belgium (477 ng m$^{-3}$, Zdrahal et al., 2002), Lumbini, Nepal (734 ng m$^{-3}$,
Wan et al., 2017) and Beijing, China (590 ng m$^{-3}$, Cheng et al., 2013). Beside levoglucosan, relative
concentration of other anhydrosugars (mannosan and galactosan) in all size-segregated aerosols were
negligible (<70 ng m$^{-3}$, not shown).
**3.5 Signature of biomass burning emissions**
Biomass primarily consists of different bio polymers (e.g. cellulose, hemicellulose, lignin,
suberin, sporopollenin and chitin) with small proportion of lipids and terpenoids. During thermal
combustion, such biomass emits different types of organic molecules, some of which has the potential
to be considered as signature molecule based on their long residence time and chemical stability
(Banerjee et al., 2015). The major combustion product of cellulose and hemicellulose includes
anhydrosugars like levoglucosan (1,6-anhydro-β-D-glucopyranose, C$_6$H$_{10}$O$_5$) and its two isomers
(mannosan and galactosan). Among these, levoglucosan is a robust and widely used tracer for biomass
burning emissions, both globally (Simoneit et al., 1999; Schkolnik et al., 2005; Cheng et al., 2013), and
over IGP (Li et al., 2014; Banerjee et al., 2015; Wan et al., 2017). In our case, levoglucosan was
abundant in submicron particles with a peak during November (week 6 to 9, Fig. 6). The rise in
concentration was universal in each particulate size fractions, but typically in submicron (837±83 ng
m$^{-3}$) and fine particulates (311±47 ng m$^{-3}$), having 54-70 % rise against rest of the monitoring period.
This could correspond to a short-term variation in emissions source strength which possibly have well
influenced the regional aerosol property. A ratio between levoglucosan with rest of the anhydrosugars





was also considered to indicate the dominating type of biomass burning, with a ratio <10 specific for
softwood combustion, and >10 for burning of hardwood and crop residues (Cheng et al., 2013). Even
a ratio >40 was reported from physical experiments using rice straw, wheat straw and maize stalks
(Engling et al., 2009). Although, the presence of mannosan and galactosan was not frequent in our
case, but an overall ratio >50 refers the exclusive dominance of agriculture residue burning across the
IGP.

7        The possibility of considering $K^+$ and $NH_4^+$ as biomass burning tracers were investigated in

terms of their association with levoglucosan for submicron and fine particulates. In general, the
temporal trend of levoglucosan coincided well with both $K^+$ and $NH_4^+$, and all these tracers registered
a gradual rise in concentration during November. Highly significant correlation ($R^2$) between
levoglucosan and $K^+$ ($PM_{1.1}$: 0.80, $PM_{1.1-2.1}$: 0.76; $p<0.01$), and levoglucosan and $NH_4^+$ ($PM_{1.1}$: 0.95, $PM_{1.1-2.1}$: 0.60; $p<0.01$) were noted at 99 % confidence interval. That definitely indicates that levoglucosan,
$K^+$ and $NH_4^+$ have similar biogenic sources over IGP which predominately contribute to the aerosol
loading, especially in $PM_{1.1}$ and $PM_{1.1-2.1}$. The relation between levoglucosan with $K^+$ and $NH_4^+$ further
appeared to be non-linear, with an exponential fit for submicron ($R^2$: 0.84, 0.94) and for fine
particulates ($R^2$: 0.83, 0.65). Non-linear correlations between levoglucosan and $K^+$ are also reported at
Amazon (Schkolnik et al., 2005) and in Beijing (Cheng et al., 2013) during extreme biomass burning
emissions. There was also evidence that $NH_4^+$ was better associated with levoglucosan compared to
$K^+$, referring the presence of additional $K^+$ sources across the region (like fireworks, Kumar et al., 2016).
However, in absence of aerosol organic carbon content, contribution of biomass burning to aerosol
mass was not computed.

22        Besides using conventional biomass burning tracers, we also evaluated the association of

submicron and fine particulate bound levoglucosan with weekly averages of Delta-C and UVAI (Fig. 6).
Both Delta-C and UVAI are the measures of identifying the relative dominance of absorbing aerosols
in the environment. In all scenarios, significant correlation ($R^2$) was noted between levoglucosan with
Delta-C (0.65, $p<0.01$) and UVAI (0.66, $p<0.01$). In addition to the ground-based aerosol measurement,
dynamic profile of trace gases concentration, especially for those that behave as aerosol precursors,
are assessed from Real-time Air Quality Data inventory of Central Pollution Control Board
(https://app.cpcbccr.com/ccr). The hourly average concentrations of individual trace gases were
initially checked for data quality and outliers, and further averaged to 24 h. No such universal trend in
concentration of all the trace gases was evident, except an overall increasing trend for NO, $NO_2$, NOx,
and CO, while $SO_2$ remained stable and there was a negative trend for $O_3$. The most striking feature in
the abundance of all the trace gases was to have an increase in concentration particularly during
November, although of different magnitude. This was also evident in the variation of particulate





bound biomass tracers, which inspire us to consider two different aerosol loading scenarios e.g.
scenario 1 for biomass burning dominating period (week 6 to 9, BDP) and scenario 2 for biomass
burning less dominant period (week 1-5 and week 10-11, BLDP). Such classification was intended to
recognize if there is any variation in aerosol source fields over IGP and in aerosol-induced radiative
forcing.
**3.6 Spatio-temporal nature of aerosol columnar properties**
Spatio-temporal variations in aerosol columnar properties and trace gases are plotted in Fig.
7a, including the daily variations at the ground station (Fig. 7b). Instead of considering the columnar
properties for the entire season, spatial plots are generated for two different scenarios like BDP and
BLDP.
The spatial pattern in aerosol columnar properties was typical having a very high aerosol
loading exclusively over IGP (area weighted AOD mean±SD: 0.55±0.21) in comparison to the rest of
South Asia (0.31±0.21). However, there was no such temporal variation particularly over IGP as both
$BDP_{AOD}$ (0.56±0.23) and $BLDP_{AOD}$ (0.53±0.23) was almost similar. The $BDP_{AOD}$ was slightly higher (12 %)
to that of reported decadal average (0.50±0.25, Kumar et al. 2018), and was comparable to the season
specific average over IGP (0.55±0.20; Kumar et al., 2018). It should be noted that area weighted AOD
average includes all the pixels retrieved across the region, some of which may not represent the
biomass emissions. This leads us to further retrieve and compare AOD particularly over the ground
station. In this case, the mean AOD was significantly high during post-monsoon (0.81±0.39), 44 %
higher for $BDP_{AOD}$ (0.98±0.42) in respect of $BLDP_{AOD}$ (0.68±0.32). Even, the $BDP_{AOD}$ was 46 % higher
compared to decadal average for the station (0.67±0.28; Kumar et al., 2018). Figure 7a also includes a
comparison of relative dominance of aerosol types in terms of AE, and in both conditions fine particles
(AE; BDP: 1.5, BLDP: 1.7) were found to dominate with a season specific mean (±1σ) of 1.6 (±0.2).
Following the evidence of persisting high AOD and high AE indicating the dominance of fine
particulates of anthropogenic origin, the nature of aerosols in terms of absorbing and/-or scattering
was distinguished through satellite observation. OMI UVAI has been widely applied to detect dust
(Badarinath et al., 2010), biomass burning aerosols (Torres et al., 2013; Kaskaoutis et al., 2014) and
soot particles (Kumar et al., 2016), and has also been used in combination with CALIPSO to detect
height of aerosol layer (Guan et al., 2010). In our experiment, the daily UVAI varied from (-) 0.34 to (+)
2.24 with a seasonal mean (±1σ) of 0.99 (±0.49) over IGP, which is considerably higher than the
seasonal mean for entire South Asia (0.47±0.46). Interestingly, negative UVAI was only evident during
early October (week 1) signifying presence of non-absorbing aerosols (like sulphate), while the
dominance of UV absorbing aerosols such as smoke and/-or mineral dust was evident during rest of



the season. During BDP, the high UVAI values (>1.5) were mainly found to concentrate over the upper
to middle IGP with 72 % of observations remain >1.0. This clearly indicates the larger abundance of
fresh UV-absorbing particles, and is similar to the reported UVAI (<2.0) over the Himalayas during peak
burning season (Kumar et al., 2011; Vadrevu et al., 2012). There was also considerable difference
between the periodical mean UVAI for BDP (1.47±0.64) and BLDP (0.75±0.58) over the ground station.
Further, following Guan et al. (2010) to use UVAI as a proxy to compute aerosol height, we found a
low average height of aerosol layer (~1.5 km), possibly due to low-altitude injection of plumes from
burning of agricultural residues.
Apart from aerosols, spatial variation of few trace gases (e.g. CO and $NO_2$), directly emitted
from biomass burning are also estimated. The MERRA-2 reanalysis surface CO profile was consistent
with the observed UVAI, with high CO surface concentration over IGP (mean±SD: 156±62 ppbv) in
comparison to South Asia (114±52 ppbv). Similar was the case for tropospheric $NO_2$ column density as
Aura OMI observations show consistent variation across IGP (2.4±1.1 x$10^{15}$ mol.cm$^{-2}$) with reference
to South Asia (1.5±1.0 x$10^{15}$ mol.cm$^{-2}$). There was definite spatial signature of the influence of biomass
emissions on these trace gases, while their abundance may also have influenced by other
anthropogenic sources (like industry and vehicular emissions). Likewise, higher surface $NO_2$
concentrations (> 5x$10^{15}$ mol cm$^2$) were particularly evident over Punjab and Delhi, over industrial
sectors in the Chhattisgarh and in lower IGP (particularly over Dhaka). Episode specific spatial
variations in mean CO (143 to 169 ppbv) and $NO_2$ concentrations (2.3 to 2.5 x$10^{15}$ mol cm$^{-2}$) were not
so radical both across IGP and over the ground station (CO: 140-142 ppbv; $NO_2$: 2.3-2.5 x$10^{15}$ mol cm$^{-}$
$^2$). The possible explanation for such minimum episode-specific variation may be the short residence
time of $NO_2$ and CO, as $NO_2$ rapidly photo-dissociate by reaction with OH radical, while CO gradually
oxidized to form $CO_2$. Overall, spatio-temporal nature of aerosol and trace gases were consistent with
the observed trend at the ground station and were prudent for establishing the influence of biomass
emissions over the region.
**3.7 Vertical distribution of aerosols**
Vertically resolved aerosol subtypes from spaceborne lidar for selected overpasses across IGP
are plotted in Fig. 8a, with corresponding extinction coefficient of aerosol type (Fig. 8b). The CALIPSO-
CALIOP profile clearly indicates a temporal change in aerosol type, without any considerable change
in the height of aerosol layer. During initial days (in October), dominance of polluted dust (dust mixed
with biomass burning smoke) were noted across IGP, with occasional prevalence of smoke (biomass
burning aerosols), clean continental (clean background aerosol) and dust aerosols. However, the
contribution of polluted dust to total aerosol extinction was higher compared to the rest of aerosol
type. The height of aerosol layer was relatively low (<1.5 km) corresponding to a low plume injection



height and thereby, pose limited potential for dispersion. The aerosol vertical profile however,
modified from the end of October due to biomass burning emissions, with dominance of smoke
particles, mainly persisting at low altitude (<1.5 km). The height of smoke layer was consistent to that
of OMI UVAI projected aerosol height. Smoke particles were found to associate with polluted dust,
clean continental and polluted continental, with overlapping profiles. Overall, smoke was the most
frequent aerosol type with high aerosol extinction coefficient (1-2.5 $Km^{-1}$ at 532 nm), and the altitude
of largest occurrence frequency of smoke remain below ~1.5 km. The low injection height of smoke
plumes from biomass burning may serve as a key input for aerosol transport modeling over IGP, as it
critically regulates the distance and direction of the particle dispersion (Guan et al., 2010; Banerjee et
al., 2011).
The daily variation in total aerosol extinction and aerosol extinction only by smoke particles
were also included in Fig. 8c. Total aerosol extinction indicates a corresponding increase during
biomass burning which peaks particularly in November, with low smoke injection height. Clear
evidence of gradual increase in smoke particle aerosol extinction was also noted. A single evidence of
high smoke extinction (>1 $Km^{-1}$) at a greater height (~ 3.4 km) was noted on November 11, which may
be associated to particles travelling from a larger distance. Overall, the CALIOP aerosol profiles were
in accordance to the ground observations and OMI UVAI, referring exclusive dominance of high UV-
absorbing aerosols across the plain during intense biomass burning.
**3.8 Potential aerosols sources and transport**
The regional transport of aerosols from western dry regions (Sen et al., 2016, 2017) and/or
from middle-east Asia (Kumar et al., 2015, 2017b, 2018) through prevailing westerlies are often
considered as a prominent contributor of aerosols across IGP. There are also reports of gradual
accumulation of fine particles at lower altitude from north-western dry regions (Kumar et al., 2015).
Considering the possibility of regional transport of biomass burning aerosols, MODIS fire counts, fire
radiative power, brightness temperature along with 5-days back trajectories were included in Fig. 9.
Active fire counts from the Terra and Aqua MODIS fires and thermal anomalies (with ≥70% confidence)
clearly indicate that fire spots were predominately over the upper IGP, mainly concentrated over the
Indian state of Punjab, Haryana and western Uttar Pradesh, and in Punjab state of Pakistan. However,
there was a temporal shift in the total number of fire counts (Fig. 9, within the marked region) from
biomass burning dominating period (BDP: 5272) to less dominating period (BLDP: 4466). Even, the Fire
Radiative Power (FRP) i.e. rate of energy released in unit time indicates a relative change in amount
and strength of biomass burning emissions, mainly during BDP (138,366 MW) in comparison to BLDP
(112,168 MW). The total FRP was higher during BDP mainly due to higher number of fire counts and
fire strength, as the rate of release of thermal radiation is related to the amount of biomass burnt and



smoke being released (Schroeder et al., 2010). The MODIS fire spots (with brightness temperature),
specially subset over IGP were plotted against five days air-mass back trajectories, simulated and
integrated at three vertical heights (100m, 300m and 500m) over the ground station. Vertical heights
were selected based on the average planetary boundary layer height (402±81 m) for the monitoring
period. The air-mass back trajectories indicate the upper IGP as the sole source of aerosols during BDP,
which was otherwise influenced by both continental and marine air-masses during non-dominating
period. The air-mass back trajectories during BDP overlap precisely on the fire spots that corresponds
to higher brightness temperature, referring greater relevance to FRP. The air masses for individual
episode were further subject to cluster and CWT analysis to quantify potential influences of aerosol
source regions to total aerosol loading (Kumar et al., 2018). The CWT were drawn considering
columnar aerosol load, and result was consistent with our prior observations. High CWT (>0.8) during
BDP was clearly attributed to the regional pollution, mainly originated from the upper IGP. In contrast,
relatively low CWT was noted during BLDP, originating both from upper IGP (CWT<0.8), western dry
region (CWT<0.6) and few from oceanic environment (CWT<0.4). This leads us to conclude with
confidence that there was a strong temporal gradient in post-monsoon specific biomass burning
emission over the upper IGP, which greatly influence the regional aerosol climatology and thereby,
possibly influence the aerosol-induced health effects and regional climate.
**3.9 Aerosol radiative forcing and atmospheric heating**
Daily satellite retrieved AOD, TCO, CWV, SSA, ground-based BC mass concentration, aerosol
water soluble and insoluble fractions were used as an input to OPAC model to simulate aerosol
radiative forcing (ARF at 0.2-4.0 μm). Within the period, TCO varied between 237 to 277 DU without
any difference between BDP (257±10 DU) and BLDP (256±12 DU). The SSA (at 550nm), designates the
fraction of scattered light over the total light extinction, was lower during BDP (0.86±0.05) compared
to BLDP (0.98±0.04), suggesting abundance of strong absorbing aerosols especially during BDP. The
CWV also fluctuates considerably (range: 0.28-3.92 cm) with overall season specific mean (±σ) of 2.0
(±0.7) cm.
The direct ARF and heating rate were estimated under clear-sky conditions with SBDART
model using OPAC output. The composite ARF was calculated for individual episodes at surface (SRF),
top of the atmosphere (TOA) and atmosphere (ATM) (Fig. 10). Overall, the ARF at TOA and SRF were
negative, indicating the aerosol cooling effect at surface and at top-of-the-atmosphere. There was a
slight temporal change in TOA radiative forcing (BDP: -28; BLDP: -23 W m$^{-2}$) compared to the
considerable intra-seasonal variation in SRF forcing (BDP: -163; BLDP: -79 W m$^{-2}$). The variation in SRF
forcing was mainly induced by the surface BC (mean; BDP, BLDP: 9, 7 μg m$^{-3}$), aerosol mass
concentration (501, 327 μg m$^{-3}$) and WSIC fractions, particularly in $SO_4^{2-}$ (38, 15 μg m$^{-3}$), $NO_3^-$ (19, 12



µg m⁻³) and NH₄⁺ (11, 4 µg m⁻³). Since the ATM forcing is the balance of attenuation of radiation at TOA
and SRF, the resultant atmospheric forcing was found very high, especially during biomass burning
dominated period (BDP: 135 W m⁻²), compared to non-dominating one (BLDP: 56 Wm⁻²). Overall, there
was a clear indication of intraseasonal variation in aerosol radiative forcing, which may be useful in
parametrization of aerosol schemes in regional climate model. Similarly, the corresponding heat rate
was substantially high during BDP (4.3 K day⁻¹), possibly influenced by more absorbing aerosols,
compared to BLDP (1.8 K day⁻¹). The computed ARF during post-monsoon was comparable to other
urban sites in Indo-Gangetic Plain that are reported to be influenced by biomass burning e.g.  Delhi
(44-131 W m⁻², Bisht et al., 2015), Patiala (57-63 W m⁻², Sharma et al. 2017), Kanpur (30-43 W m⁻²,
Kaskaoutis et al., 2013) and over Karachi (35-84 W m⁻², Alam et al., 2011). However, none of the earlier
reports noted the intraseasonality in ARF by means of change in driving factors which, appeared to be
significant, and necessitate proper addressing in regional model simulation. Intraseasonality in ARF
was earlier reported over Varanasi during winter (ARF: 31-47 W m⁻², Kumar et al., 2017b), while the
change in forcing was not as drastic as evident during post-monsoon. Therefore, it is extremely likely
that intraseasonality in aerosol properties significantly influence the aerosol-climate-health
interactions over IGP and therefore, must need to be taken in to account for uncertainty analysis in
the regional aerosol/-climate model.
**4. Conclusions**
The influence of biomass burning emissions on aerosol properties, transport and radiative
forcing was evaluated over Indo-Gangetic plain, South Asia. Very high concentration of total and fine
mode aerosol (PM₂.₁) were observed during post-monsoon, with significant increase in fine to coarse
particle ratio (>1) particularly from November. Submicron particles dominate the aerosol fine mode,
with PM₁.₁ to PM₁.₁₋₂.₁ ratio frequently exceeding 2.5. The WSIS was found to constitute greater
proportion of submicron and fine particle mass compared to the coarser one. The WSIS was mainly of
secondary nature, with major contribution from sulfate and nitrate ions. A strong correlation between
NH₄⁺ and SO₄²⁻, and high NH₄⁺/SO₄²⁻ equivalent ratio in submicron particulates indicate the abundance
of gaseous NH₃ to neutralize acidic species (SO₄²⁻). This contrasted with coarse mode particles where
low NH₄⁺/SO₄²⁻ equivalent ratio refers the predominant neutralization by crustal minerals. The NO₃⁻ to
SO₄²⁻ ratio for submicron and fine mode particles also increased (>1) during extreme biomass
emissions, as expected considering other reported observations of haze events over Asia. A rise in
black carbon with corresponding increase in Delta-C refer to the added contribution of biomass
burning emissions. The influence of emissions was further quantified using specific molecular
(Levoglucosan), inorganic (K⁺ and NH₄⁺) and satellite (UVAI) tracers. Levoglucosan was the most
abundant species in submicron particles, with a very high ratio (>50) against other anhydrosugars





denoting exclusive emissions from burning of agriculture residues. The temporal variation in
levoglucosan was consistent with inorganic tracers ($K^+$ and $NH_4^+$), with a sharp rise during November,
and a strong correlation between these three indicates their biogenic sources. The association
between levoglucosan and $K^+$ or $NH_4^+$ was non-linear, with an exponential fit for submicron and fine
particulates. The spatio-temporal distribution of aerosols was evaluated in terms of area weighted
mean both over IGP and over the selected transect across ground station. During biomass burning
dominated period, a considerable increase in columnar aerosol loading was highlighted (AOD: 0.98),
consisting absorbing aerosols (UVAI > 1.5) with a corresponding low plume height (~1.5 km).
Moreover, the variation of few trace gases associated with biomass emissions (CO and $NO_2$) were
consistent with AOD, allowing a definite spatial signature of emissions sources and transport across
IGP. The CALIPSO-CALIOP cross-sectional altitudinal profiles clearly illustrate the intraseasonality in
aerosol types that were dominated by smoke and polluted continental aerosols during biomass
emissions, which otherwise associate to clean continental, polluted dust and dust aerosols. The
possible pathway for regional transport of aerosols from upper IGP to the ground station was noted
using cluster analysis and concentration weighted air mass back-trajectories. Finally, aerosol optical
and micro-physical properties were used in combination to simulate direct aerosol radiative forcing
(ARF) and atmospheric heating. There was evidence of strong intraseasonality in ARF with very high
atmospheric forcing (135 $Wm^{-2}$) and heating rate (4.3 $Kday^{-1}$) during biomass burning dominated
period compared to non-dominating one (56 $Wm^{-2}$, 1.8 $Kday^{-1}$).

20          Considering that the duration of these biomass burning emissions due to post-monsoon

specific agricultural practices represents several weeks per year, there annual impact on ARF and by
consequent on the regional climate is not negligible. We therefore, conclude with reasonable level of
confidence that intraseasonality in aerosol properties must be seriously considered in the regional
aerosol-climate model, for improve assessment and forecasting of aerosol-climate-health interactions
across IGP.
**Data availability**

27          MODIS data are available at Level 1 Atmosphere Archive & Distribution System (LAADS) at

https://ladsweb.nascom.nasa.gov. Aura-OMI and MERRA 2 reanalysis data are available at Mirador-
NASA    Goddard    Earth    Sciences    Data    and    Information    Center    (GES    DISC)
(https://mirador.gsfc.nasa.gov). CALIPSO data are available at NASA Atmospheric Science Data Center
(https://eosweb.larc.nasa.gov). Planetary Boundary Layer height and air mass back-trajectories are
retrieved from Global Data Assimilation System (GDAS) archives hosted at NOAA-Air Resource
Laboratory (https://ready.arl.noaa.gov). Modis Fire products are obtained from Fire Information for
Resource Management System (FIRMS) (https://firms.modaps.eosdis.nasa.gov). Trace gases data at





ground station are available at Real time Air Quality Data inventory of Central Pollution Control Board
(https://app.cpcbccr.com/ccr).
**Team List.** Nandita Singh (NS), Tirthankar Banerjee (TB), Made P. Raju (MPR), Karine Deboudt (KD),
Meytar Sorek-Hamer (MSH), Ram S. Singh (RSS) and Rajesh K. Mall (RKM).
**Author Contributions**
N.S. and T.B. designed the experiment while N.S., M.P.R. and T.B. carried out the experiment and
analyzed the data. N.S., M.P.R., K.D., T.B., R.S.S., R.K.M. and M.S.H. interpreted the observation and
N.S., T.B. and K.D. drafted the manuscript.
**Competing interests.** The authors declare that they have no conflict of interest.
**Acknowledgements**
The research is supported by Science and Engineering Research Board (SERB), Department of Science
and Technology (DST), New Delhi (SR/FTP/ES-52/2014). T.B. acknowledges the financial support from
University Grants Commission (UGC) under UGC-ISF bilateral project (6-11/2018 IC), R.S.S.
acknowledges the Indian Space Research Organization (ISRO), Thiruvananthapuram under ARFI (Code:
P-32-13) and R.K.M. acknowledges DST under Prime Ministers National Action Plan on Climate Change
project (DST/CCP/CoE/80/2017-G). N.S. acknowledges the financial support under DST Women
Scientist scheme (SR/WOS-A/EA-1012/2015) and M.S.H. acknowledges the NASA Post-Doctoral
Fellowship, administered by USRA. Authors duly acknowledge the guidance and cooperation provided
by Dean and Director, IESD-BHU.

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





**List of figures:**
Fig. 1. Geographical location of aerosol ground monitoring station (a), and MODIS aerosol optical
depth with NCEP/NCAR composite means of wind vector during monitoring period (b).
Fig. 2. Time series of (a) size segregated particulate mass concentration, (b) particle ratio and (c) daily
means of meteorological variables.
Fig. 3. Variation of (a) ions and (b) trace metals in different aerosol size fractions.
Fig. 4. Variation of BC, Delta C and ABL during entire monitoring period.
Fig. 5. Size-segregated particulate bound (a) organic aerosols, and difference in the molecular
compositions of (b) *n*-alkanes and (b) fatty acids.
Fig. 6. Temporal variation of trace gases and biomass burning signature molecules ($NH_4^+$, $K^+$,
Levoglucosan), and their associations within different aerosol size fractions.
Fig. 7. Episode specific spatial distribution of AOD, AE, UVAI, surface CO (ppbv) and tropospheric $NO_2$
(molecules $cm^{-2}$) over (a) South Asia and (b) at ground station.
Fig. 8. Aerosol vertical profiles from selected CALIPSO overpasses across ground station (a) aerosol
subtypes, (b) extinction coefficients of each aerosol type and (c) time series of extinction profile
for total and smoke aerosols.
Fig. 9. Episode specific MODIS fire count, fire radiative power (FRP, MW), brightness temperature (B.
Temp., K), and five days air mass back-trajectory along with CWT.
Fig. 10. Episode specific aerosol short wave radiative forcing and atmospheric heating.



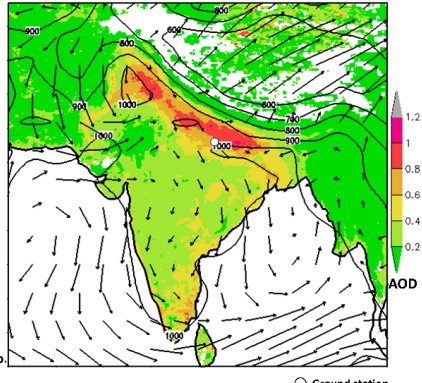

Fig. 1. Geographical location of aerosol ground monitoring station (a), and MODIS aerosol optical depth with NCEP/NCAR composite means of wind vector during monitoring period (b).

**Note**: Background image in (a) was retrieved from Suomi NPP VIIRS satellite indicating the thick aerosol layer over north India on October 31, 2016.





3    Fig. 2. Time series of (a) size segregated particulate mass concentration, (b) particle ratio and (c) daily
4        means of meteorological variables.

5    **Note**: Week 1 to 5 are in the month of October, week 6 to 9 are in November and week 10 to 11 are
6        in December.



4  Fig. 3. Variation of (a) ions and (b) trace metals in different aerosol size fractions.

5  **Note**: Week 1 to 5 are in the month of October, week 6 to 9 are in November and week 10 to 11 are

6  in December.



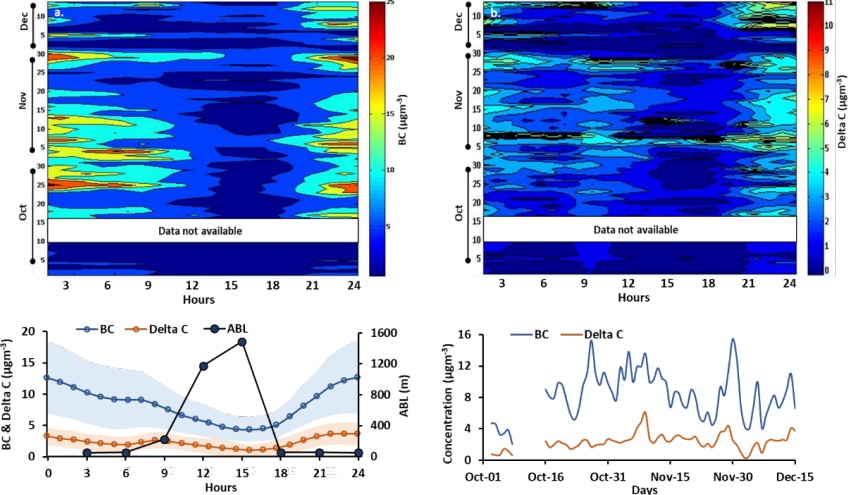

5        Fig. 4. Variation of BC, Delta C and ABL during entire monitoring period.

6     **Note**. The blue and red shade in the graph at lower panel indicates the standard deviation.







3
4    Fig. 5. Size-segregated particulate bound (a) organic aerosols, and difference in the molecular
5           compositions of (b) *n*-alkanes and (b) fatty acids.



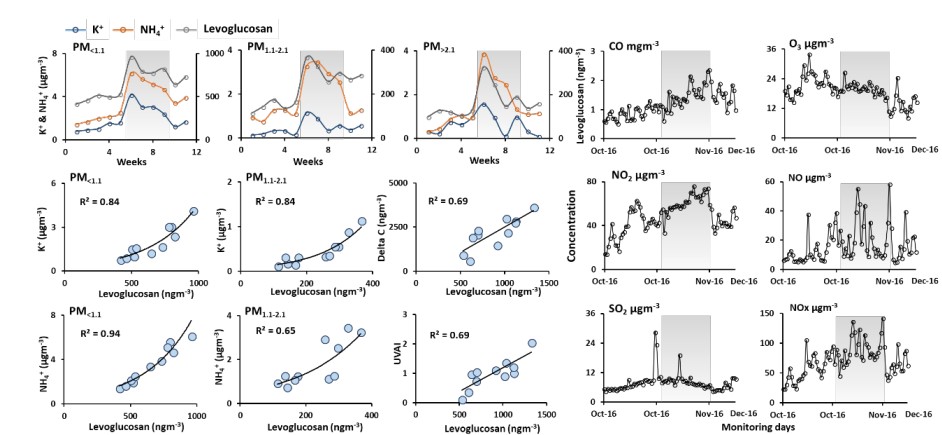

4   Fig. 6. Temporal variation of trace gases and biomass burning signature molecules ($NH_4^+$, $K^+$,
5      Levoglucosan), and their associations within different aerosol size fractions.

6   **Note.** The shaded area indicates the peak biomass burning emissions.





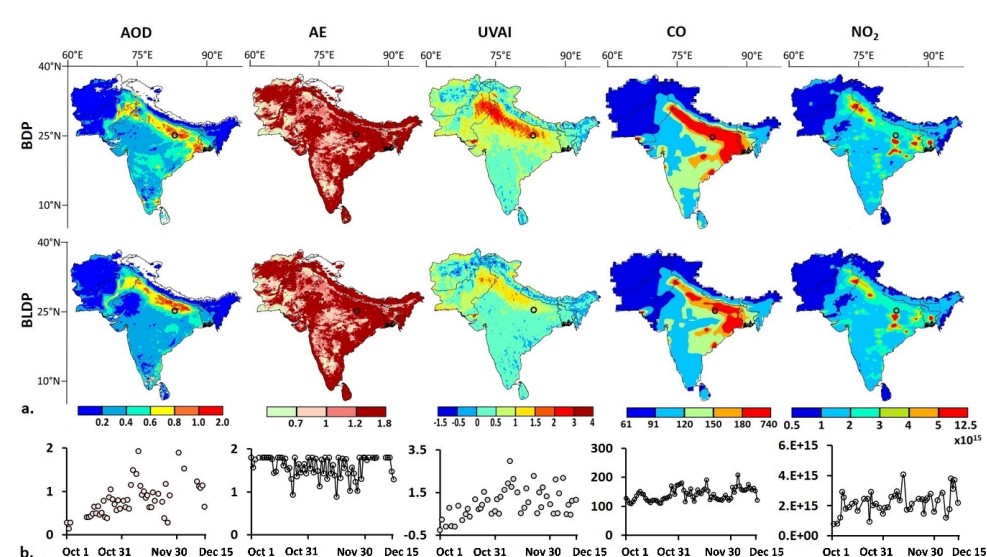

Fig. 7. Episode specific spatial distribution of AOD, AE, UVAI, surface CO (ppbv) and tropospheric $NO_2$
(molecules $cm^{-2}$) over (a) South Asia and (b) at ground station.

**Note**. The lower panel indicates the time-series for each parameter retrieved particularly over the
ground station.





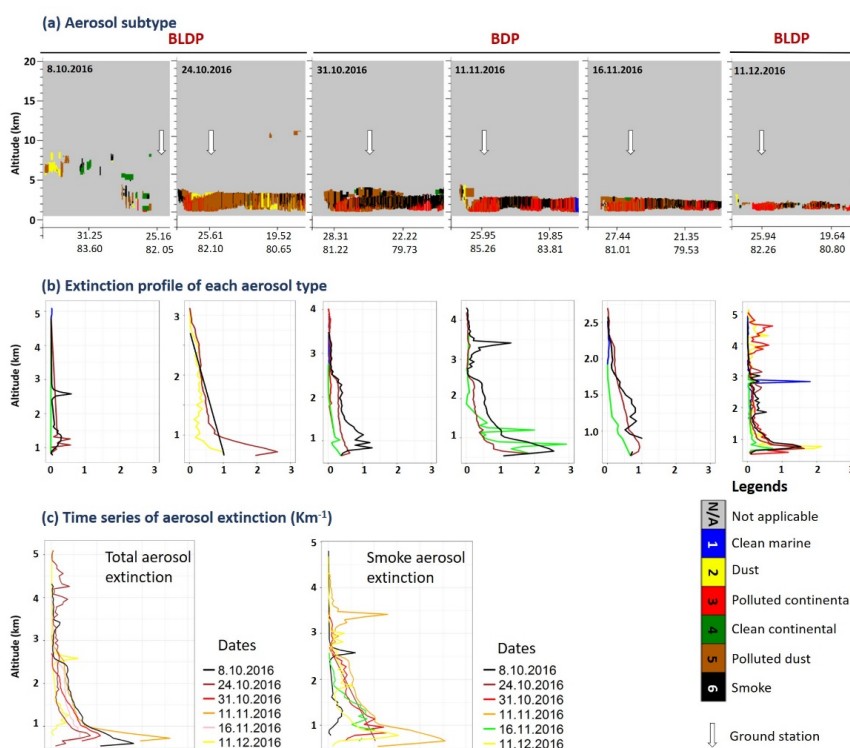

4    Fig. 8. Aerosol vertical profiles from selected CALIPSO overpasses across ground station (a) aerosol
5         subtypes, (b) extinction coefficients of each aerosol type and (c) time series of extinction
6         profile for total and smoke aerosols.



Fig. 9. Episode specific MODIS fire count, fire radiative power (FRP, MW), brightness temperature (B. Temp., K), and five days air mass back-trajectory along with CWT.





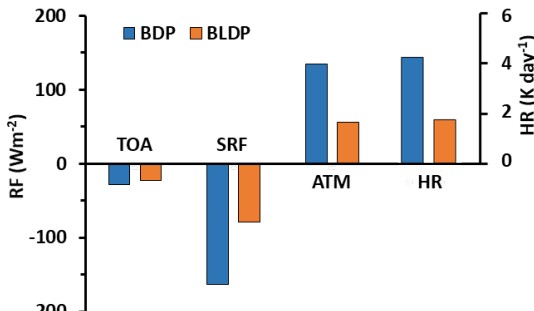

3
4       Fig. 10. Episode specific aerosol short wave radiative forcing and atmospheric heating.

