# Peer review of "Manuscript under review for journal Atmos. Chem. Phys."

_Atmospheric Chemistry and Physics, 2018_

## Referee Comment (RC1) · Anonymous Referee #1 · 14 Jul 2018

Singh et al. analyse the effects of biomass burning on aerosol distribution, chemistry, and radiative forcing over the Indo-Gangetic Plain combining in situ and satellite-based observations and radiative transfer calculations. Manuscript can be considered for publication in ACP however several comments should be addressed.

Page 1, l.36: "weighted of air trajectories" to "weighted air trajectories"

Page 1, l.41: "must need to be studied" to "are needed"

Page 1, l.42: "in much finer scale to improve parameterization of aerosol/-climate model across the region." This is not clear. Rewrite or remove.

[Figure]

Page 4, l.10: wunderground.com data is validated with regional weather monitoring station data! Why is that needed, and how good the validation results turned out to be?

Page 4, l.13: What is meant by simulated meteorological observations?

Page 4, l.14: "measure" to "analyze" in context of NCEP data.

Page 8, l.12: "OPAC derived outputs were tuned in respect to measured relative humidity". This is not clear. OPAC outputs are for different humidity ranges. How could output be tuned further! clarify / rewrite

Page 8, l.15-16: "as an input" to "as inputs"

Fig.8: Top row, Y scale should have been 0-10 km or so, there is not much data seen above that altitude

Section 3.3.: key sources of BC aerosols over the stations and nearby should be discussed based on literature (Kumar et al., JGR, 2015)

Page 13, l.23: "has" to "have"

---

## Referee Comment (RC2) · Anonymous Referee #2 · 14 Jul 2018

Comments on the manuscript by Singh et al. (acp-2018-446)

In this work, the authors carried out a quite comprehensive research on the biomass burning emissions during post-monsoon in South Asia (Indo-Gangetic Plain), involving aerosol composition, transport and radiative forcing. The topic is interesting and important. However, currently the manuscript has some critical problems. The relevant discussion appears just be piled up and superficial. For example, the results of mass concentration, BC, ions, levoglucosan, as well as satellite remote sensing and source areas are already well known in this region. Each subtopic mentioned above actually has already been presented in the literatures. So the authors need to point out what is the new finding from this work. Otherwise, it will undermine the novelty of this study. Specific comments:

1. Page 5, Line 22-26, here the authors did not mention Ca, K, Na in the analysis, although they are presented in Figure 3 and related discussions. So how did you measure these major elements? For the trace elements, the information of data quality control is also lack. It is well known that the quartz filters have high blank values for some trace elements.

   Actually, according to the discussion (In section 3.2.2), the contents regarding trace elements is not closely related to the theme of this work (i.e. biomass burning). So I suggest to delete this part.

2. For the organic compounds, similarly, I can not judge the quality of the analysis in this work. What is the recovery, accuracy or precision of the organic compounds?

3. Page 12, Line 15-16, reference is needed here. And it's better to give more explanation.

4. Line 31-32. Yes, PAHs is important for the study of emissions from biomass/fossil combustion. However, if you can not give proper interpretation of PAHs results, I suggest to delete it.

5. In this work, many items (organic tracers, major ions) were determined in the

laboratories. However, organic carbon and elemental carbon (OC/EC) was not included. Obviously, it is very vital to interpret the results of organic tracers combined with OC/EC, considering the focus of this work is biomass burning.

6. Section 3.5, here levoglucosan was introduced in details. Actually it already been presented in section 3.4. (Page 12, Line 33). So some changes are needed for a better logic.

7. Page 13, Line 23, here you mean the ratio is L/(M+G)?

8. Page 14, Line 8, it is common to see potassium occurs in crustal minerals

9. Page 16, Line 12. I do not think so. The variations of CO and NO2 shown in Figure 7b did not reflect the influence of intensive biomass burning.

---

## Author Comment (AC1) · 28 Aug 2018

Title: Aerosol chemistry, transport and climatic implications during extreme biomass burning emissions over Indo-Gangetic Plain

MS No.: acp-2018-446

Authors sincerely appreciate the careful reviews and suggestions provided by the reviewer and thank the reviewer and the Editor for their time to evaluate the manuscript. Authors have made appropriate changes to the manuscript in response to the comments that have considerably improved the manuscript. In authors' response, authors have responded point-by-point to comments (reviewer comments in *blue*, authors' responses in *black*), and have included the revisions in the text with and without tracked-changes.

**Authors' Responses to Referee # 1**

Singh et al. analyse the effects of biomass burning on aerosol distribution, chemistry, and radiative forcing over the Indo-Gangetic Plain combining in situ and satellite-based observations and radiative transfer calculations. Manuscript can be considered for publication in ACP however several comments should be addressed.

1.  Page 1, l.36: "weighted of air trajectories" to "weighted air trajectories"
    Modified in the revised text.

2.  Page 1, l.41: "must need to be studied" to "are needed"
    The sentence has been modified (page 1, l.39-41).

3.  Page 1, l.42: "in much finer scale to improve parameterization of aerosol/-climate model across the region." This is not clear. Rewrite or remove.
    Authors emphasized that such detailed characterization of aerosol chemistry over IGP will be useful for reducing uncertainties in regional aerosol-climate model. However, as suggested, authors have modified the text (page 1, l.39-41).

4.  Page 4, l.10: wunderground.com data is validated with regional weather monitoring station data! Why is that needed, and how good the validation results turned out to be?
    To understand the implications of meteorology on particulate mass, daily mean of meteorological variables was required. The regional weather monitoring station, as maintained by India Meteorological Department (IMD), is although located close to the particulate sampling station however, only reports daily maximum (at 1730 h) and daily minimum (at 0830 h). To be accurate, authors have considered daily means from wunderground.com (WU), which reports weather data collected directly from automated weather stations operating at airports (here in Babatpur, Varanasi). The aerial distance of Varanasi airport to particulate monitoring station is 23 km. We therefore, compared daily maximum and minimum (as reported by IMD) against WU reported observations and found no significant difference. Likewise, WU reported daily maximum ($R^2$: 0.955) and minimum temperature ($R^2$: 0.964) was found well validated against IMD reported observations.

5.  Page 4, l.13: What is meant by simulated meteorological observations?
    This was in context of ABL height (at 0.5°) which was retrieved from NCEP's Global Data Assimilation System (GDAS). The GDAS is the system used by the NCEP Global Forecast System (GFS) model to place observations from individual station into a gridded model. GDAS adds meteorological observations like surface observations, balloon data, wind profiler data, aircraft reports etc. from a station to simulate a gridded, 3-D, model space available at various resolutions.

6.  Page 4, l.14: "measure" to "analyze" in context of NCEP data.

Modified in the revised text (Page 4, l.14).

7. Page 8, l.12: "OPAC derived outputs were tuned in respect to measured relative humidity". This is not clear. OPAC outputs are for different humidity ranges. How could output be tuned further! clarify / rewrite.
Authors admit the error and deleted the text from the revised manuscript (Page 8, l.14). Authors wish to mention that the OPAC derived outputs (AOD and SSA) are reconstructed in a way so the modelled (OPAC output) and observed/satellite derived values matches within ±5% deviation. Average relative humidity for dominating period (RH: 70%) and for non-dominating period (RH: 80%) was however, considered separately as the prevailing RH only to simulate OPAC model.

8. Page 8, l.15-16: "as an input" to "as inputs"
Modified in the revised text.

9. Fig. 8: Top row, Y scale should have been 0-10 km or so, there is not much data seen above that altitude.
Figure 8 has been modified accordingly.

10. Section 3.3.: key sources of BC aerosols over the stations and nearby should be discussed based on literature (Kumar et al., JGR, 2015)
Thanks for the suggestion. Authors have addressed the point that diurnal variation in BC was in fact not driven by anthropogenic emissions rather by the changes in the regional meteorology, especially ABL. Authors have included additional discussion on BC sources in the revised text based on Kumar et al. 2015b (Page 11-12, l.29-4).

11. Page 13, l.23: "has" to "have"
Modified in the revised text.

---

## Author Comment (AC2) · 28 Aug 2018

Title: Aerosol chemistry, transport and climatic implications during extreme biomass burning emissions over Indo-Gangetic Plain MS No.: acp-2018-446

Authors sincerely appreciate the careful reviews and suggestions provided by the reviewer and thank the reviewer and the Editor for their time to evaluate the manuscript. Authors have made appropriate changes to the manuscript in response to the comments that have considerably improved the manuscript. In the authors' response, authors have responded point-by-point to comments (reviewer comments in *blue*, authors' responses in *black*), and have included the revisions in the text with and without tracked-changes.

**Authors' Responses to Referee # 2**

In this work, the authors carried out a quite comprehensive research on the biomass burning emissions during postmonsoon in South Asia (Indo-Gangetic Plain), involving aerosol composition, transport and radiative forcing. The topic is interesting and important. However, currently the manuscript has some critical problems. The relevant discussion appears just be piled up and superficial. For example, the results of mass concentration, BC, ions, levoglucosan, as well as satellite remote sensing and source areas are already well known in this region. Each subtopic mentioned above actually has already been presented in the literatures. So the authors need to point out what is the new finding from this work. Otherwise, it will undermine the novelty of this study.

Authors highly appreciate such constructive comments and have addressed these issues in the revised manuscript. To authors knowledge, there was no published report available till date on influence of biomass burning on air borne particulate over IGP, measured considering size-segregated particulates (PM1.1, PM1.1-2.1, PM>2.1) and submicron (PM1) particulate chemistry and using satellite data to assess the spatial nature of pollution. Previous reports mainly used PM2.5 or TSP (total aerosols) as matrices to assess emission budget (Rajput et al., 2014), organic mass-to-organic carbon ratio (Rajput and Sarin, 2014), emissions of PAHs (Rajput et al., 2011), organic molecular tracers (Wan et al., 2017) and radiative forcing (Sharma et al., 2017; Alam et al., 2011); while only few have explored remote sensing observations to interpret fire (Vadrevu et al., 2012). Considering completely diverse physicochemical properties of submicron and coarser particulates, our analysis was novel especially in terms of:

- 1. Integrating satellite & ground-based observations to assess impact over the ground-station and across IGP.
- 2. First report considering size-segregated aerosols (PM1.1, PM1.1-2.1 and PM>2.1) with detail aerosol chemistry for PM1.1.
- 3. First report on PM1.1 bound PAHs and organics tracers like Levoglucosan during biomass burning emissions.
- 4. We have also reported spatial and vertical distribution of air pollutants & its short-term variations across IGP.
- 5. First report on time-series of total and smoke aerosol extinction profile during biomass burning emissions.

These novel aspects of the manuscript have been addressed in introduction (page 3, I.6-27).

**Specific comments:**

1. Page 5, Line 22-26, here the authors did not mention Ca, K, Na in the analysis, although they are presented in Figure 3 and related discussions. So how did you measure these major elements? For the trace elements, the information of data quality control is also lack. It is well known that the quartz filters have high blank values for some trace elements.

Authors admit there was a mistake and in the revised text Ca, K, Na were added in the methodology (section 2.3.3, page 5, I.25). These metals were also analyzed by AAS along with other trace metals.

Yes, authors agree with reviewer's point that the quartz filters have high blank values for some trace metals. However, their levels were very low in comparison to ambient samples. For metal analysis, the blank filter papers (unexposed quartz filters) were treated and analyzed similarly like real ambient samples. The measured trace metal concentrations in the blank samples were further deducted from the metal concentration from ambient samples to have metal concentration in ambient air.

**Actually, according to the discussion (In section 3.2.2), the contents regarding trace elements is not closely related to the theme of this work (i.e. biomass burning). So I suggest to delete this part.**

Authors are thankful for such constructive comments. There are evidences of trace metal emissions from burning of biomass, especially in PM1. Likewise, Wang et al. (2015) have concluded biomass combustion as the most prominent source of Fe concentration in submicron particles. For global emission estimation of Fe, Wang et al. (2015) showed combustion as predominant emission source of Fe over Indo-Gangetic plain in comparison to dust. Beside Fe, there are also reports of trace metals emissions particularly K, Cu, S, Zn, Pb from burning of rice-straw (Ryu et al., 2012); organic bound Fe2+, Cu2+, Ni2+, Zn2+ from hardwood burning (Graham et al., 2011) and Cu, Pb, Ni, As from the burning of biomass fuel (Zhang 2014).

Considering these evidences, authors have included a detail discussion (Page 11, I. 6-23) on submicron (PM1.1) and PM1.1-2.1 bound metals in the manuscript. Briefly, a massive increase in Fe (59-415%) and in K (119-528%) concentration is reported for submicron and fine aerosols during biomass burning period.

2. For the organic compounds, similarly, I can not judge the quality of the analysis in this work. What is the recovery, accuracy or precision of the organic compounds?

Authors are thankful for reviewers' suggestion to improve the QA/QC of analytical procedure. Authors wish to state that we have performed the routine recovery test of organic compounds before the sample analysis and now this has been included in the revised manuscript (Page 6, l.11-14). The recoveries of organic compounds were tested by spiking the known concentration of standard compounds on the pre-combusted quartz filters. They were extracted and analyzed in identical to the real samples. The average recoveries and respective RSD (in parenthesis) of the n-alkanes (28 compounds) ranged from 72-92% (1-12%), phthalates (6 compounds) ranged from 75-88% (2-7%), FAMES ranged from 74-92% (1-9%), PAHs ranged from 73-93% (1-10%) and anhydrosugars (3 compounds) ranged from 75-80% (4-6%, data may be shared if required). To improve the clarity, we have incorporated the recovery of organic compounds and RSD in the revised text (Page 6, l.11-14).

**3. Page 12, Line 15-16, reference is needed here. And it's better to give more explanation.**

Authors have modified the section 3.3 with additional justifications to the BC sources considering relevant references like Kumar et al., 2015b; Wang et al., 2011; Kumar et al., 2016. Authors have addressed that the diurnal variation in BC concentration was primarily influenced by the changes in the regional meteorology, especially ABL (Page 11-12, I.29-4). In contrast, for daily variation in BC, there was a clear influence of additional anthropogenic emissions like biomass burning during November. This was established with the increase in Delta-C that represent smoke emissions from biomass burning (Wang et al., 2011; Kumar et al., 2016).

**4. Line 31-32. Yes, PAHs is important for the study of emissions from biomass/fossil combustion. However, if you can not give proper interpretation of PAHs results, I suggest to delete it.**

To our knowledge, this is the first report of submicron particulate bound PAHs during extensive biomass burning period over IGP. There were only few efforts to characterize the PAHs in PM2.5 and TSP bound aerosols across IGP for biomass burning emissions (like by Chen et al., 2015; Rajput et al., 2011). In revised text authors have strengthen the discussions on PM1.1 and PM1.1-2.1 bound PAHs considering all the relevant references which have accounted the biomass burning emissions (Page 13, I.4-11).

5. In this work, many items (organic tracers, major ions) were determined in the laboratories. However, organic carbon and elemental carbon (OC/EC) was not included. Obviously, it is very vital to interpret the results of organic tracers combined with OC/EC, considering the focus of this work is biomass burning.

Authors agree that the consideration of OC/EC would have additionally strengthen the discussions on organic tracers. We wished to included EC/OC measurement, but our quartz samples were limited in terms of particulate exposure. We have used Non-Viable Anderson Cascade Impactor (Tisch, USA) for particulate sampling which gives the deposition of aerosol particles in the form of dots (dia. 1mm or less), scattered on filter disc. For EC/OC analysis, the measurement assumes that aerosol particles are uniformly deposited on filter disc and concentrations are measured in terms of unit area. However, it would have not possible to measure the area of each dot and accurately quantify the EC/OC concentration for cascade samples.

6. Section 3.5, here levoglucosan was introduced in details. Actually it already been presented in section 3.4. (Page 12, Line 33). So some changes are needed for a better logic.

In section 3.4, authors emphasized only on characterizing organic compounds in size-segregated aerosols and levoglucosan concentration was discussed with reference to other available literature.

However, in section 3.5, the emphasis was solely to establish relationship of levoglucosan with other established biomass burning markers, to find out the type of biomass burning and their short-term variations. So, the perspective was different for discussions related to levoglucosan and its isomers in section 3.4 and 3.5.

**7. Page 13, Line 23, here you mean the ratio is L/(M+G)?**

Authors believe that the reviewer wished to indicate the term 'ratio' cited in line 3 of page 14. Yes, the ratio is in between levoglucosan (L) and sum of mannosan (M) and galactosan (G). We have modified the text for clarity (Page 14, I.3).

**8. Page 14, Line 8, it is common to see potassium occurs in crustal minerals**

Authors acknowledge the reviewers' comment that K commonly occurs in crustal minerals, but this holds generally true for coarser particulates ( $PM_{>2.5}$ ). As per our understanding, the crustal materials are mostly found in coarser particles and K+ should have considered from crustal origin if we found elevated K+ in  $PM_{>2.5}$ . In literature (Banerjee et al., 2015; Chen et al. 2017 and references therein), K+ concentration in finer particles is well reported of biomass burning origin. Even, we evaluated the association in between levoglucosan and K+ for all three size fractions and found highly significant correlation only in submicron ( $PM_{<1.1}$ ) followed by fine range particles ( $PM_{1.1-2.1}$ ), referring origin of K+ mainly from biomass burning emissions.

**9. Page 16, Line 12. I do not think so. The variations of CO and NO2 shown in Figure 7b did not reflect the influence of intensive biomass burning.**

Authors have modified the argument in page 16, line 14-17 and conclude that increase in CO and NO2 profile over IGP is the possible consequences of anthropogenic emissions (including industrial, vehicular and biomass burning emissions), not solely due to the biomass burning.